# Implant or Tooth?—A Prospective Clinical Study on Oral Health-Related Quality of Life for Patients with “Unrestorable” Teeth

**DOI:** 10.3390/jcm11247496

**Published:** 2022-12-17

**Authors:** Maria Bruhnke, Michael Naumann, Florian Beuer, Insa Herklotz, Mats Wernfried Heinrich Böse, Stefan Neumeyer, Manja von Stein-Lausnitz

**Affiliations:** 1Department of Prosthodontics, Geriatric Dentistry and Craniomandibular Disorders, Charité—Universitätsmedizin Berlin, Corporate Member of Freie Universität Berlin, Humboldt—Universität zu Berlin and Berlin Institute of Health, Aßmannshauser Straße 4-6, 14197 Berlin, Germany; 2Private Practice Dr. Neumeyer & Partner, Leminger Straße 10, 93458 Eschlkam, Germany

**Keywords:** forced orthodontic extrusion, endodontically treated tooth, oral health-related quality of life, patient perception, forced eruption, orthodontic extrusion, implant therapy, implant-supported single crown, unrestorable tooth, severely damaged tooth, patient-related outcome measures

## Abstract

In cases of severely compromised teeth, dental practitioners are confronted with the therapeutic decision of whether to restore a tooth or replace it with an implant. Comparative scientific evidence on patient perception of both treatment approaches is scarce. The subject of this prospective clinical study was to compare oral health-related quality of life (OHRQoL) between two treatment groups: restoration of severely destroyed teeth after orthodontic extrusion (FOE) versus tooth extraction and implant-supported single crown restoration (ISC). A self-selected trial was performed with 21 patients per group. OHRQoL was assessed with the aid of the Oral Health Impact Profile (OHIP-G49) at different time intervals: before treatment (T1), after treatment (T2), after restoration (T3) and at recall (T4). Overall, OHIP scores improved from baseline to follow-up for both concepts with no significant differences between groups. There were no significant differences in subscales between FOE and ISC at T1, T3 and T4. In terms of functional limitations (*p* = 0.003) and physical disability (*p* = 0.021), patients in the FOE group temporarily exhibited lower OHRQoL at T2 in comparison to the ISC group. However, at baseline, after final restoration and at recall, the study demonstrates the same level of OHRQoL for both treatment concepts.

## 1. Introduction

Extensively damaged teeth are often classified as “unrestorable” due to their defect extension and location. Subgingival and subcrestal defect margins are problematic as they usually are accompanied by a violation of biologic width [1,2]. Moreover, for long-term success of postendodontic restorations, a sufficient ferrule design after preparation is necessary [3,4]. Therefore, in cases of tooth preservation, pre-prosthetic treatment measures such as crown lengthening [5] or orthodontic extrusion procedures [6,7] are deemed necessary. As surgical crown lengthening may be associated with interproximal bone loss and an inevitable lengthening of the clinical crown [8], this treatment modality may be disadvantageous, especially in the anterior region. In contrast, forced orthodontic extrusion is regarded as a less invasive treatment procedure, maintaining hard and soft tissues [9]. However, scientific evidence is limited to case reports [6,10] and case series [11,12].

Therefore, tooth restoration after pre-prosthetic treatment measures with consecutive endodontic therapy and post-and-core restoration as opposed to extraction and implant placement are therapy options that must be carefully considered and discussed [13]. As tooth extraction is usually followed by soft and hard tissue volume loss [14], in cases of implant placement, often extensive augmentation procedures are deemed necessary. Therefore, implant placement, especially in the esthetic anterior region, is classified as a highly complex procedure [15]. In order to avoid augmentation, immediate implant placement is gaining popularity, but remains controversially discussed [16,17,18]. Clinical studies have shown that buccal mucosal recession is a common problem that may occur within the first years after treatment [19,20].

As no uniform treatment recommendation for severely compromised teeth exists, dental practitioners choose different treatment options [21]. Thus, in general, the choice of treatment is influenced by education and clinical expertise [22].

Both treatment concepts, implant placement after extraction and tooth restoration after forced orthodontic extrusion, are suited for the comparable clinical situation of a severely damaged, “unrestorable” tooth. We are unaware of studies comparing both treatment concepts in regard to survival and success rates with a similar observation time. Moreover, a direct comparison between treatment modalities regarding patient-related outcomes and evidence on economic data is largely absent.

The Oral Health Impact Profile is a standardized questionnaire to compare patient-related outcomes and was first described in 1994 by Slade and Spencer [23]. It was validated over the past decades in multiple versions and countries and is regarded as an appropriate instrument to measure patient-related outcomes during dental treatment procedures [24]. The questionnaire includes 49 items for analysis of patient condition.

Therefore, the aim of the study was to compare oral health-related quality of life during treatment and at recall in patients receiving implant-supported single crowns (ISC) and patients with severely damaged teeth restored after forced orthodontic extrusion (FOE). The null hypothesis tested was that there would be no significant difference in oral health-related quality of life between both treatment groups.

## 2. Materials and Methods

### 2.1. Experimental Design

A prospective study design was selected to compare oral health-related quality of life for patients undergoing forced orthodontic extrusion procedures (FOE) and implant therapy (ISC) in a self-selected clinical trial. Ethical approval was obtained by the Ethical Committee of Charité—Universitätsmedizin Berlin, Germany (application number: EA2/300/20). The study was registered at DRKS (German Clinical Trials Register, registration number: DRKS00030342). Between January 2020 and October 2021, study participants were recruited from a group of consecutive patients at the Department of Prosthodontics, Geriatric Dentistry and Craniomandibular Disorders of the Charité (Berlin, Germany). All FOE procedures were supervised by the author M.B., and all ISC procedures by authors M.W.H.B. and I.H. under standardized protocols. All patients provided written informed consent forms. Sample size calculation was based on data from a clinical comparative study [25] assuming a scattering of data with 0.16. With *n* = 20 patients per group, the *t*-test has a power of 80% with an effect size of 0.145 and a two-sided significance level of 0.05. Sample size calculation was performed using the procedure MTT0-1 (*t*-test for two means) in nQuery 8.1.0.0.

A total of 42 patients 18 years and older were assessed for eligibility for both treatment groups. For treatments with ISCs, the following inclusion criteria were defined: (1) single tooth gaps; (2) completion of surgical, conservative and periodontal pretreatment; and (3) class I or II according to risk classification of American Society of Anesthesiologist (ASA). Exclusion criteria were: (1) systematic diseases that preclude implant surgery; (2) chemo- or radiation therapy; (3) chronic inflammation, systematic diseases and metabolic disorders in association with bone lesions; (4) ASA risk classification exceeding class II; (5) medication affecting bone remodeling; and (6) fresh extraction sockets. 

For FOE treatments, the following inclusion criteria had to be met: (1) participants in need of treatment of a severely destroyed tooth with 2 proximal contacts and an intact dentition; (2) tooth mobility ≤ 1; (3) crown-root ratio after treatment ≤ 1; (4) defects violating the biologic width or missing ferrule design preparation; and (5) prospective single tooth restoration. Teeth with vertical root fractures, ankylosis or hypercementosis, tooth mobility and probing depths ≥ 3 mm at defect location were excluded. Participants had to be willing to appear for follow-up appointments and be able to maintain proper oral hygiene in both treatment groups.

### 2.2. Interventions

#### 2.2.1. Implant-Supported Single Crown Treatment (ISC)

Figure 1, Figure 2, Figure 3 and Figure 4 demonstrate the clinical workflow for implant-supported single crowns (ISC). Before treatment, for all participants, conventional gypsum casts were obtained for implant planning. For each tooth to be replaced, a conventional wax-up served as a reference for the prospective restoration. For backward planning, the models and the wax-up were scanned using a laboratory scanner (D2000, 3Shape, A/S, Copenhagen, Denmark). Data were exported as STL datasets and superimposed with DICOM data from CBCT (Veraviewpocs 3D R100, J. Morita Europe GmbH, Dietzenbach, Germany) for each participant in two types of implant planning software: SMOP version 2.13 (SwissmedaAG, Baar, Switzerland) for CAMLOG SCREW-LINE implants (C-SL, CAMLOG Vertriebs GmbH, Wimsheim, Germany) and CoDiagnostiX (Dental Wings GmbH, Chemnitz, Germany) for Straumann Bone Level Tapered implants (S-BLT, Straumann gmbH, Freiburg, Germany). Stereolithography technique was used for the fabrication of surgical guides. Implant placement followed a strict protocol according to the manufacturer’s instructions. After 3–6 months, all implants were restored with screw-retained single crowns.

#### 2.2.2. Forced Orthodontic Extrusion Treatment (FOE)

Figure 5, Figure 6, Figure 7 and Figure 8 show the clinical workflow for forced orthodontic extrusion procedures (FOE). Conventional gypsum models were obtained from each patient to measure the interocclusal space for a monodirectional extrusion. Probing depths, defect size and radiographic images were analyzed to determine the amount of extrusion and the prospective crown-to-root ratio for each participant. After removal of carious decay and insufficient restorations, a fiber-reinforced composite-based bar (Extrusion pin, Komet Dental, Lemgo, Germany) was adhesively luted on the root surface of the tooth to be extruded with a self-adhesive resin cement (RelyX Unicem 2 Automix, 3M Deutschland GmbH, Neuss, Germany). A second bar was placed on adjacent teeth with flowable composite (Tetric EvoFlow, Ivoclar Vivandent AG, Schaan, Liechtenstein). Elastics initiated occlusal movement of the tooth, and patients changed the elastics twice a day. Supracrestal fibrectomy and scaling and root planning procedures were carried out at the same appointment [26]. Details on clinical workflow are described elsewhere in detail [12]. After extrusion, teeth were bonded to adjacent teeth with composite resin (Tetric EvoCeram, Ivoclar AG) for retention of at least two months. After the retention time, glass-fiber posts (X-Post; Dentsply Sirona, Charlotte, NC, USA) were used where indicated. All teeth were restored with single crowns.

### 2.3. Assessment of Oral Health-Related Quality of Life (OHRQoL)

Outcome was assessed using OHIP (Oral Health Impact Profile). Each OHIP question includes a five-point ordinal rating scale: “never” = 0, “hardly ever” = 1, “occasionally” = 2, “fairly often” = 3, “very often” = 4. Oral health-related quality of life is assessed as the sum of 49 OHIP items (range: 0 to 196 points). Furthermore, items were grouped in subscales indicating different impairments of oral health-related quality of life: functional limitations, physical pain, psychological discomfort, physical disability, psychological disability, social disability and handicap. Question items 47, 48 and 49 were discarded as they refer to removable dentures. Lower scores indicate better health-related quality of life and higher scores indicate poorer oral health-related quality of life. The OHIP-G49 was used four times at similar stages of therapy for both treatment groups at follow-up after treatment: for FOE before treatment (T1 = baseline), after retention (T2), after restoration (T3) and at recall (T4), and accordingly for ISC before treatment (T1 = baseline), after implant placement (T2), after restoration (T3) and at recall (T4). Recalls were performed 12 months after restoration for both treatment groups. Patients completed the questionnaires unassisted. However, the operator was present in case of questions by participants. The primary outcome was the comparison of the total sum scores and subscales for OHIP-G49 in both treatment groups at different time levels (T1–T4). Medians were calculated for the overall, FOE and ISC total sum scores for T1 to T4. Statistical analysis was performed using the Mann–Whitney U Test for independent samples with a statistical software program (IBM SPSS Statistics, v25; IBM Corp., Armonk, NY, USA). The level of significance was set at 5%. For 92% of questionnaires (*n* = 151), there were no missing values. In all questionnaires with less than 10% of missing information, data were imputed using the patient’s mean score (*n* = 12) [27]. For questionnaires with more than 20 missing items, data were excluded from further analyses (*n* = 1).

## 3. Results

### 3.1. Baseline Characteristics

In this prospective study, a total of 42 participants with a mean age of 47 ± 19 years for FOE (forced orthodontic extrusion) and with a mean age of 51 ± 15 years for ISC (implant-supported single crowns) were assessed for eligibility and included in this self-selected clinical trial. Table 1 summarizes patient and tooth characteristics. For FOE, 47% of teeth were incisors or canines, 48% were premolars and one tooth was a molar. For ISC, most implants were premolars and molars (96%). One patient was not available for T3 and subsequent follow-ups due to personal circumstances in the ISC group. One patient aborted FOE treatment at T2. The overall drop-out rate was 5%.

### 3.2. Oral Health-Related Quality of Life

The baseline OHIP median total sum score was 39 points at T1 (Table 2). For FOE before treatment, the median OHIP total sum score was 33, and for ISC, it was 43.5 points, with no significant differences between groups (*p* = 0.927). After initial treatment (T2), oral health-related quality of life decreased in the FOE group, indicated by an increase of 11 to a median of 42 points. OHRQoL improved in the ISC group at T2, reflected in a decrease in OHIP scores to a median of 27 points (Figure 9). The difference was not statistically significant (*p* = 0.059). After restoration (T3), a decrease in OHIP scores to a median of 18 for FOE and to 17 for ISC was recorded. At recall (T4), the median OHIP score decreased to 10.5 for both treatment groups. Statistical analysis showed no significant differences between groups.

Table 3 summarizes the median sum scores for subscales of OHIP-G49 in both treatment concepts. There were no significant differences in subscales between FOE and ISC at T1, T3 and T4. At T2, after initial treatment, patients in the FOE group exhibited poorer OHRQoL in comparison to patients in the ISC group in regard to functional limitations (*p* = 0.003) and physical disability (*p* = 0.021). For subscales of physical pain, psychological discomfort, psychosocial disability, social disability and handicap, there were no significant differences between groups.

## 4. Discussion

The present prospective clinical trial is the first study reporting on patient-related outcome measures (PROMs) after treatment of deeply destroyed or “unrestorable” teeth with forced orthodontic extrusion (FOE) in comparison to extraction of respective teeth and implant-supported restorations (ISC). This clinical study compared oral health-related quality of life (OHRQoL) in patients receiving either tooth-based or implant-borne restorations. In regard to overall OHIP median scores, it was found that OHRQoL did not differ significantly between treatment groups FOE and ISC at time intervals T1 (baseline), T2 (after treatment), T3 (after restoration) and T4 (at recall). Therefore, the null hypothesis of the present study is partially accepted. However, subscale data show that patients in the FOE group exhibited poorer OHRQoL in comparison to patients in the ISC group initially after treatment at T2 regarding functional limitations and physical disability, which is important and unexpected (in contrast to surgical procedures of implant placement) regarding patient perception.

Higher median OHIP sum scores for FOE may possibly be explained by the nature of this therapy method in terms of treatment time and effort. Time of orthodontic extrusion as well as time of retention, possible necessity for periodontal surgery, endodontic treatment and possible retreatment, as well as post-and-core restoration all prolong the treatment process. Furthermore, poorer OHRQoL due to short-term functional limitations and physical disabilities may be explained by the orthodontic appliance itself, since it may limit function in terms of chewing ability or lead to physiological discomfort due to local strains of elastics. Additionally, a reactive unilateral chewing pattern and difficulties in changing the elastics may occur. Moreover, esthetic appearance may be impaired and be accompanied by pronunciation difficulties. However, this conclusion should be interpreted with caution as tooth characteristics are not evenly distributed between groups. In the ISC group, the majority of teeth were molars and premolars, while only one single canine tooth was included. In contrast, for the FOE group, half of treated teeth were in the anterior region and and only one molar was included. Therefore, a distinct differentiation in regard to tooth location and its impact on results is not possible, and future research should address this issue.

As tooth extraction is usually accompanied by hard and soft tissue loss [14], hard and soft tissue augmentation measures are deemed necessary. The fact that complex augmentation procedures were excluded from data analysis is one of the major limitations of this clinical study, as these procedures may also affect OHRQoL. Patients presented with different initial clinical situations, e.g., most teeth had already been extracted in the ISC group, while in the FOE group, teeth were still in situ. Therefore, patients in the ISC group did not have to wait until healing after tooth extraction and were ready to accept implant surgery. This circumstance may have affected the results of the present study.

Overall, median sum scores of this clinical trial are comparable to another clinical study comparing OHRQoL of patients two years after endodontic treatment with dental implant therapy [28]. The results are in accordance with our study as the authors demonstrated high satisfaction for both treatment modalities. However, patients exhibited more physical pain during endodontic treatment in contrast to implant therapy. In the present study, OHIP was assessed with the aid of the G-49 questionnaire, since it is the most frequently used instrument for objective evaluation of PROMs in clinical studies. It is simple, reliable and reproducible [27,29].

Scientific evidence on forced orthodontic extrusion is limited to case reports and case series, and long-term results are not available in the scientific literature. The only clinical study on forced orthodontic extrusion reports minor orthodontic relapse in three and limited root resorption in six out of thirty-two patients after one year of observation [11]. In contrast, for implant-supported single crowns, long-term data exist and suggest an increase in survival rates from 92.6% to 97.2% when comparing old with new publications [30]. However, the incidence of biologic and technical complications seems to remain consistent.

Despite all of these aspects, the implications of these two treatment modalities may differ in clinical routine as further prognostic, anatomical and patient-related factors play a significant role in the treatment planning process [31]. For example, clinical conditions may not allow implant placement due to insufficient space to adjacent anatomic structures. The same is true for orthodontic extrusion: the interocclusal dimension should allow for a monodirectional orthodontic movement, and root anatomies, furcations or teeth with ankylosis involvement may prevent tooth movement [9]. Moreover, a sufficient root length is necessary for a favorable prospective crown-to-root ratio after treatment. However, orthodontic extrusion may be regarded at least as a valid treatment alternative for patients with absolute contraindications for surgical interventions or implant therapy [32,33], to avoid high therapy costs [34] and particularly in young, still growing patients, as age still limits the possibility of implant placement [35,36]. Despite insignificant, short-term poorer OHRQoL regarding functional limitations and physical disability with FOE during treatment, it has to be considered as a valid therapy option. Moreover, the present study provides data that can be used to better inform patients about the advantages and disadvantages of both treatment options.

FOE may be considered as first-line therapy, since in case of extrusion failure, a subsequent implant therapy is still possible. Orthodontic extrusion may even not only maintain, but also improve surrounding hard and soft tissues (site development), enabling delayed or even immediate implant placement and avoiding excessive hard- and soft-tissue augmentation procedures [16,17,18].

## 5. Conclusions

At baseline, time of restoration and recall, patients with severely compromised teeth treated with forced orthodontic extrusion (FOE) exhibit the same level of oral health-related quality of life (OHRQoL) as patients with implant-supported restorations. In terms of functional limitations and physical disability, OHRQoL of patients who underwent FOE reduced only temporarily compared to implant therapy.

## Figures and Tables

**Figure 1 jcm-11-07496-f001:**
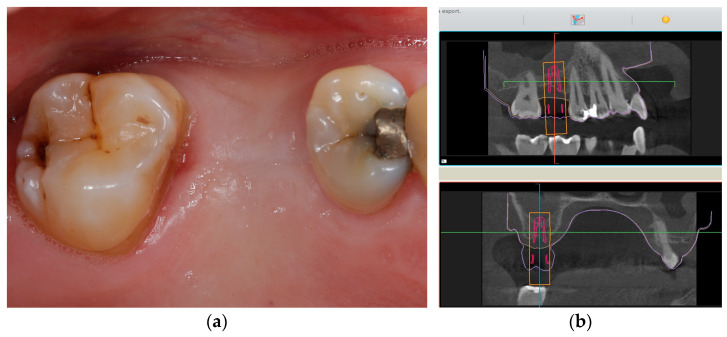
Initial clinical situation at baseline. (**a**) Initial situation from occlusal aspect before surgery. (**b**) Digital implant planning with the aid of SMOP (SwissmedaAG, Baar, Switzerland).

**Figure 2 jcm-11-07496-f002:**
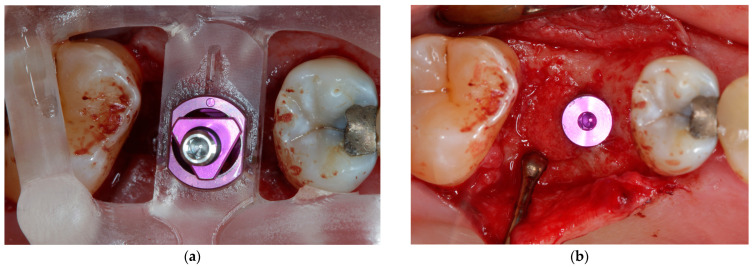
Implant placement: (**a**) implant placement through fully guided surgical guide; (**b**) inserted implant (Camlog Screw Line Promote Plus 4.3 × 9 mm).

**Figure 3 jcm-11-07496-f003:**
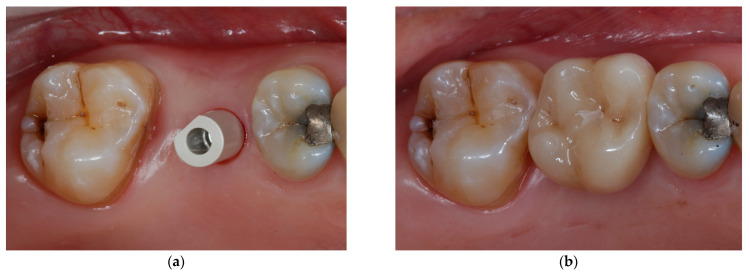
Final restoration: (**a**) digital impression taking; (**b**) final restoration with a screw-retained single crown.

**Figure 4 jcm-11-07496-f004:**
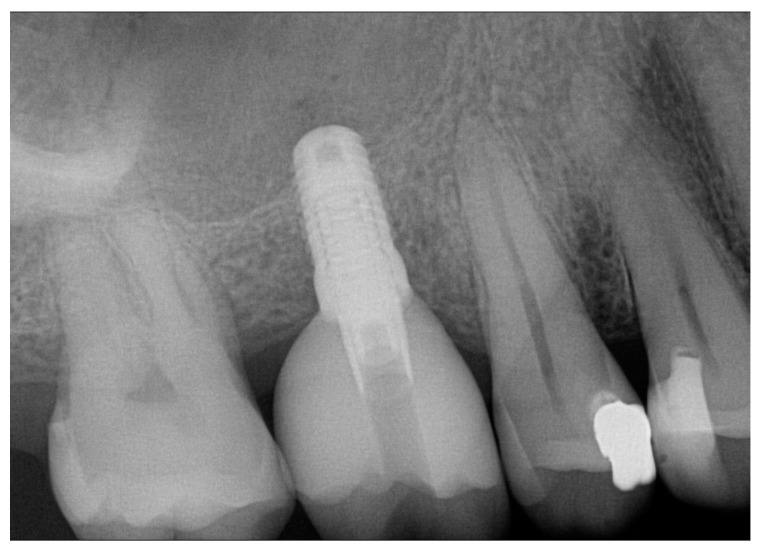
Final radiologic control shows no complications.

**Figure 5 jcm-11-07496-f005:**
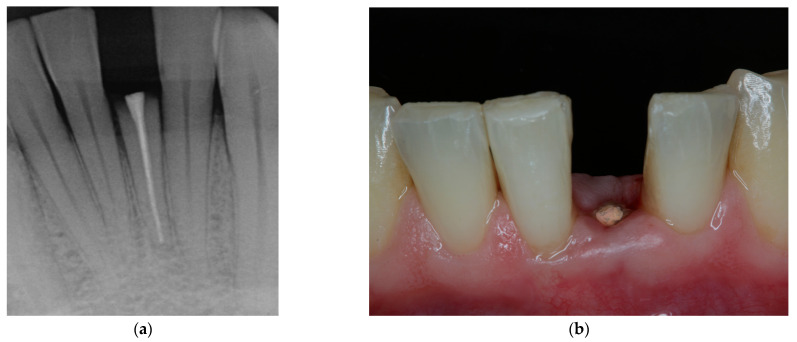
Initial clinical situation showing extensively damaged, anterior tooth in lower jaw. (**a**) Radiographic image at baseline. (**b**) View from buccal aspect shows subgingival defect extension. Restoration without pre-prosthetic treatment measures is not possible as ferrule design may not be achieved and biologic width is violated.

**Figure 6 jcm-11-07496-f006:**
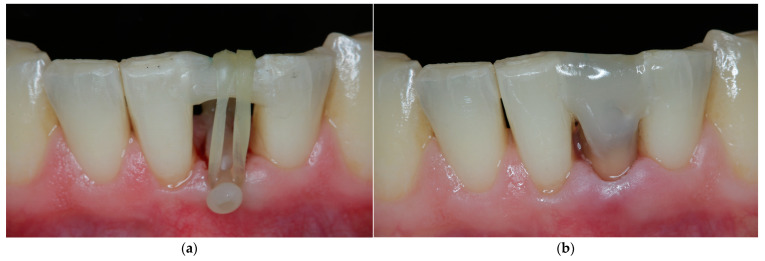
Forced orthodontic extrusion. (**a**) Construction of the orthodontic appliance: elastics initiate tooth movement in occlusal direction. (**b**) After extrusion, tooth is splinted to adjacent teeth for retention.

**Figure 7 jcm-11-07496-f007:**
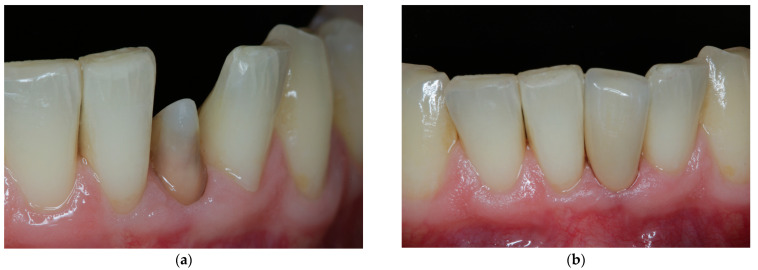
Final restoration. (**a**) Preparation from buccal aspect. (**b**) Final restoration in situ.

**Figure 8 jcm-11-07496-f008:**
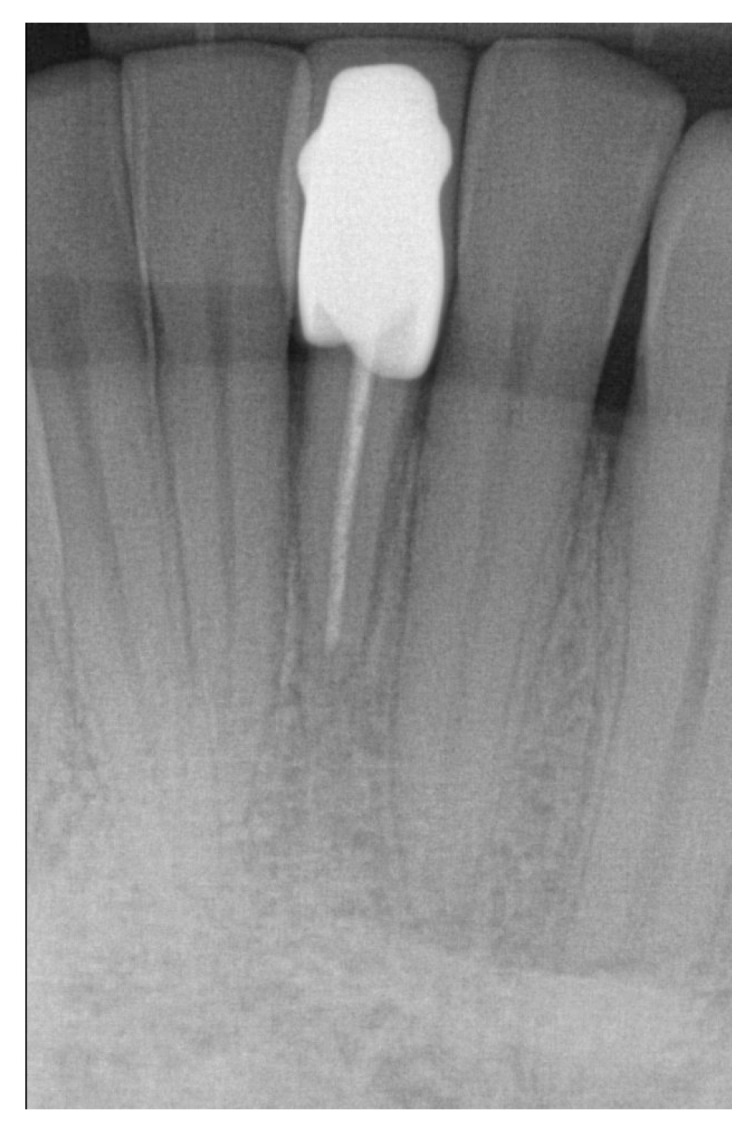
Radiologic image at recall shows stable marginal bone level and no apical translucencies.

**Figure 9 jcm-11-07496-f009:**
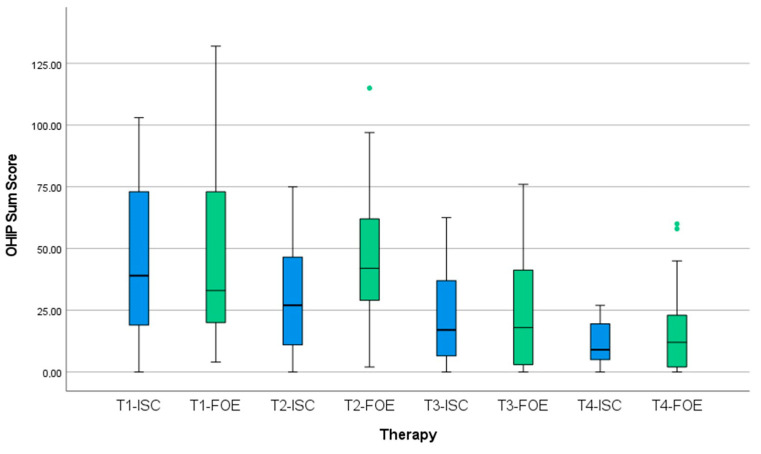
Development of oral health-related quality of life (OHRQoL) over treatment intervals (T1 = baseline, T2 = after treatment, T3 = after restoration, T4 = at recall) dependent on treatment group: forced orthodontic extrusion (FOE) versus implant-supported single crown (ISC).

**Table 1 jcm-11-07496-t001:** Descriptive data for participants and tooth characteristics.

Patient, Tooth and Treatment Characteristics		
	Forced Orthodontic Extrusion	Implant-Supported Single Crown
No. of patients	21	21
Age (y, mean ±SD ^1^)	47 ± 19	51 ± 15
Gender		
female (% of group)	5 (24%)	11 (52%)
male (% of group)	16 (76%)	10 (48%)
Tooth Location		
Incisor	7 (33%)	0
Canine	3 (14%)	1 (5%)
Premolar	10 (48%)	8 (39%)
Molar	1 (5%)	12 (57%)
Jaw Distribution		
Maxilla	16 (76%)	11 (52%)
Mandible	5 (24%)	10 (48%)

^1^ SD = standard deviation.

**Table 2 jcm-11-07496-t002:** Oral health-related quality of life (OHRQoL) for different time intervals stratified for treatment groups of forced orthodontic extrusion (FOE) and implant-supported single crowns (ISC).

Assessment	Overall		Treatment Group				*p* Value
			Forced Orthodontic Extrusion (FOE)		Implant-Supported Single Crown (ISC)		
	N	Median (Min/Max)	N	Median (Min/Max)	N	Median (Min/Max)	
T1	41	39 (0/132)	21	33 (4/132)	20	44 (0/103)	0.927
T2	42	36 (0/115)	21	42 (2/115)	21	27 (0/75)	0.059
T3	40	18 (0/76)	20	18 (0/76)	20	17 (0/63)	0.968
T4	40	11 (0/60)	20	11 (0/60)	20	11 (0/45)	0.820

T1 = at baseline, T2 = after treatment, T3 = after restoration, T4 = at recall. There were no significant differences between treatment groups.

**Table 3 jcm-11-07496-t003:** Oral health-related quality of life (OHRQoL) in mean OHIP dimensions at time intervals T1–T4 for both treatment groups.

Assessment	Overall	Treatment Group		*p*-Value
		Forced Orthodontic Extrusion (FOE)	Implant-Supported Single Crown (ISC)	
	Median (Min/Max)	Median (Min/Max)	Median (Min/Max)	
T1				
Functional limitations	8 (0/21)	8 (0/19)	9 (0/21)	0.794
Physical pain	9 (0/27)	9 (0/27)	10 (0/22)	0.754
Psychological discomfort	7 (0/20)	7 (0/20)	7 (0/18)	0.990
Physical disability	4 (0/22)	5 (0/22)	3 (0/14)	0.538
Psychological disability	5 (0/19)	5 (0/19)	6 (0/15)	0.618
Social disability	3 (0/13)	3 (0/13)	3 (0/9)	0.808
Handicap	4 (0/20)	4 (0/20)	6 (0/13)	0.843
T2				
Functional limitations	6 (0/15)	9 (0/15)	5 (0/12)	**0.003**
Physical pain	8 (0/22)	9 (0/22)	6 (0/14)	0.054
Psychological discomfort	5 (0/16)	5 (0/16)	5 (0/14)	0.503
Physical disability	6 (0/21)	7 (0/21)	2 (0/13)	**0.021**
Psychological disability	5 (0/16)	5 (0/16)	4 (0/12)	0.415
Social disability	2 (0/14)	2 (0/14)	0 (0/9)	0.312
Handicap	3 (0/12)	4 (0/12)	3 (0/10)	0.436
T3				
Functional limitations	4 (0/16)	18 (0/76)	4 (0/13)	0.620
Physical pain	5 (0/17)	5 (0/17)	5 (0/12)	0.841
Psychological discomfort	3 (0/11)	3 (0/11)	3 (0/9)	0.602
Physical disability	2 (0/15)	2 (0/14)	2 (0/15)	0.841
Psychological disability	2 (0/10)	2 (0/10)	3 (0/9)	0.820
Social disability	0 (0/8)	0 (0/8)	0 (0/5)	0.565
Handicap	2 (0/8)	1.5 (0/7)	2 (0/8)	0.640
T4				
Functional limitations	3 (0/10)	3 (0/10)	3 (0/8)	0.841
Physical pain	3 (0/17)	4 (0/17)	3 (0/15)	0.820
Psychological discomfort	1 (0/9)	1 (0/9)	1 (0/8)	0.862
Physical disability	0 (0/9)	0 (0/9)	0 (0/6)	0.583
Psychological disability	1 (0/10)	0 (0/10)	1 (0/8)	0.799
Social disability	0 (0/6)	0 (0/6)	0 (0/5)	0.841
Handicap	0 (0/8)	0 (0/8)	0 (0/7)	0.620

T1 = at baseline, T2 = after treatment, T3 = after restoration, T4 = at recall. Bold *p* values indicate significant differences between treatment groups.

## Data Availability

Not applicable.

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
