# Peer review of "Implant or Tooth?—A Prospective Clinical Study on Oral Health-Related Quality of Life for Patients with “Unrestorable” Teeth"

_jcm, 2022, doi:10.3390/jcm11247496_

Round 1

Reviewer 1 Report

The presented study tried to compare the OHRQoL of clinical procedures and results between ISC and orthodontic extrusion. However, the study design had a huge flaw. As the authors already mentioned, in the ISC group, the tooth was already extracted, therefore, the site was ready for implant surgery. The patient ddi not have to wait the healing after tooth extraction. Probably, the patients were ready to accept implant surgery compared to adjacent tooth preparation and fixed dental prostheses. Probably, a better study design would begin with the same situation, i.e. a unrestorable tooth (probably an anterior tooth). Patients may choose the treatment modality after comprehensive understanding of both treatment modalities. 

As the author mentioned in the last part of discussion, orthodontic extrusion is a powerful treatment modality in many clinical situations, however, its indication is quite limited to achieve a long-term success. Compared to orthodontic extrusion, implant therapy has little limitation to apply (I’m not talking about severely destroyed ridges situations), however, clinicians should be very careful in the esthetic region. Longterm data showed gingival recession around poorly managed anterior implant restorations. The last recall was 12 month after completion of the each treatment. I think 12 month is too short to evaluate orthodontic extrusion. The authors should add reference about the time of last recall for orthodontic extrusion. 

Reviewer 2 Report

The introduction has provided adequate background information regarding forced orthodontic extrusion. However, there is a lack of scientific evidence and clarification relating to implant placement treatment modality. Also, due to the different implications and scenarios for treatment between restoration after forced orthodontic extrusion and implant placement with a single crown, the need and meaning in clinical practice when comparing these two methods (the aim of the study) has not been clearly explained.

In terms of relevant references, most of them in the introduction has been appropriately used, except for the cite reference 13.

-       The manuscript mentioned that “implants were not compared to severely damaged but endodontically treated teeth”. In the reference 13, there was a comparison between implant-supported single crown and root canal treatment and restoration. The severity of endodontically treated teeth in the systematic review had not been clearly stated.

-       The aforementioned statement has an insignificant impact on clarifying the aim of the study, as no information on forced orthodontic treatment was noticed.

Are all the cited references relevant to the research?

Most of the references were appropriately applied throughout the study, except for the references 13, 22, and 23:

-       Inappropriate problems regarding the reference 13 have been explained above.

-       For the references 22 and 23, the results in those scientific studies were referred to as “comparable” to the result of this study. However, the OHIP sum scores in the studies 22 and 23 were applied in different subjects (with different dental treatments), not exclusively for restoration after orthodontic extrusion and implant-supported single crowns. Therefore, the comparison might not be accurate.

Is the research design appropriate?

-       As mentioned above, the comparison between implant-supported single crowns treatment and restoration after forced orthodontic extrusion treatment has not been proven necessary to provide a meaningful application in the clinical practice, as fundamentally, the implications of these two treatment modalities are dissimilar (for instance, forced orthodontic extrusion could be carried out only if the remaining tooth root has a sufficient length).

-       There is a lack of consistency regarding subjects in the two groups, thereby suggesting that the comparison is irrelevant:

·       The conditions of subjects before treatment were highly contrasting (for example, patients with ISCs had already lost teeth, while their counterparts had severely destroyed teeth with 2 proximal contacts and an intact dentition, leading to the inevitably different treatment modalities).

·       The ISC group were concerned with mostly molars and premolars, while anterior teeth were largely involved in the FOE group, and molars were considered as one of the exclusion criteria in this group.

Are the methods adequately described?

Apart from some important problems mentioned in the study design, overall, the methods section in manuscript has been sufficiently demonstrated. Nevertheless, the specific dates for conducting the study have not been mentioned.

Are the results clearly presented?

The results have been clearly demonstrated.

Are the conclusions supported by the results?

The conclusions are supported properly by the results.

English language and style

-       The writing format, including lexical resources, grammatical range and accuracy, collocation and the use of words, is formal and suitable to the standard of a scientific manuscript.

-       There is a lack of important linking words to ensure proper coherence and cohesion, especially in the Introduction and Discussion section.

Furthermore, one of the major shortcomings in the study has been mentioned in the Discussion, which is the inconsistency regarding subjects between two groups. Therefore, the reason why poorer OHRQoL regarding functional limitations and physical disabilities were seen for the FOE group might be inaccurate. Also, the topic which the study is concerned with does not demonstrate the novelty in dentistry nowadays.

Reviewer 3 Report

Manuscript of considerable interest for the dental sector, before proceeding to the evaluation for a possible publication it needs an important revision

Abstract highlight more the results obtained

Keyword; add the specs, these are few and generic

Introduction: add all the minimally invasive systems to restore the dental elements before the prosthetic products already studied by the research group of Prof. Butera et all

Materials and methods: but the sample size is missing, remaining well described

Very confusing results, they highlight the most significant data to be usable by readers less experienced in scientific works

Discussion, to add as future objectives the use of a gel with postbiotics, already studied by the research group of Prof. Scribante, in order to reduce the discomfort and increase the healing processes by reducing the chemical pharmacological action.

Conclusion, reformulate them based on the changes

Bibliography add mandatory references

Round 2

Reviewer 2 Report

-       the comparison between implant-supported single crowns treatment and restoration after forced orthodontic extrusion treatment has not been proven necessary to provide a meaningful application in the clinical practice, as fundamentally, the implications of these two treatment modalities are dissimilar (for instance, forced orthodontic extrusion could be carried out only if the remaining tooth root has a sufficient length).

-      It is not reasonable to compare the two groups above

Reviewer 3 Report

The manuscript has been revised in all its parts

You only need to change the reference 26, that of Dr. Cossellu the correct one is the following 

10.4103/2278-0203.197392

And add the use of gels with postbiotics also in implant maintenance, studied by the same research group

10.3390/app12062800
